# Comparative Analysis of Gut Microbiota Diversity Across Different Digestive Tract Sites in Ningxiang Pigs

**DOI:** 10.3390/ani15070936

**Published:** 2025-03-25

**Authors:** Wangchang Li, Xianglin Zeng, Lu Wang, Lanmei Yin, Qiye Wang, Huansheng Yang

**Affiliations:** 1Hunan Provincial Key Laboratory of Animal Intestinal Function and Regulation, Hunan International Joint Laboratory of Animal Intestinal Ecology and Health, Laboratory of Animal Nutrition and Human Health, College of Life Sciences, Hunan Normal University, Changsha 410081, China; liwangchang1019@163.com (W.L.);; 2Yuelushan Laboratory, Changsha 410128, China

**Keywords:** Ningxiang pig, 16s rRNA, gut, microbiome

## Abstract

We studied the gut microbiota of Ningxiang pigs, a protected Chinese livestock breed, focusing on four key nutrient absorption sites (gastric, ileum, cecum, and colon). Using 16S rRNA sequencing and bioinformatics analyses, we found significant differences in microbial diversity (alpha and beta diversity) among these tissues. *Firmicutes* dominated the microbiota, with *Bacteroidota* being prominent in specific regions. LEfSe analysis identified tissue-specific microbial communities, such as f_*Prevotellaceae* in the cecum and o_*Lactobacillales* in the stomach. Functional profiling revealed roles in digestion, metabolism, and cellular processes. These results highlight the importance of tissue-specific microbiota in nutrient absorption and health, offering insights for future nutritional studies.

## 1. Introduction

Pork is the primary source of meat in the daily diet of Chinese residents and has long dominated meat consumption in China [1]. It is significantly influenced by Chinese dietary culture and exhibits characteristics of rigid demand. However, issues such as rising feed ingredient prices, low feed utilization rates, and difficulties in biosafety control also restrict the development of pork production [2,3]. Therefore, improving feed digestibility, reducing waste, and lowering costs are key research directions in modern pig farming. Thus, gaining a deep understanding of specific processes in nutrient absorption during pig nutrition uptake becomes crucial, as it allows for the rational adjustment of nutritional strategies based on the characteristics of nutrient absorption.

Ningxiang pig is one of the four famous pig breeds in China, native to Ningxiang City in Hunan Province [4]. It is a typical fat-type pig breed. However, compared with commercial pigs such as Duroc × Landrace × Yorkshire (hybrid pigs), Ningxiang pigs have a lower slaughter rate, slower growth rate, and longer rearing cycle, resulting in relatively low short-term economic benefits [5]. This makes it particularly necessary to study the characteristics of nutrient absorption in Ningxiang pigs.

The gut microbiota has a profound impact on animal nutrition digestion, and certain microbial groups are closely linked to growth performance [6]. With its vast diversity and complex metabolic activities, the gut microbiota plays a key role in numerous host physiological processes, including the promotion of amino acid absorption and metabolism, maintenance of intestinal barrier integrity, and enhancement of intestinal immune function [7,8]. Studies have shown that Prevotella and Bacteroides are key microbes linking insulin resistance to branched-chain amino acid biosynthesis [9]. In addition, the gut microbiota plays a vital role in maintaining intestinal health by strengthening intercellular tight junctions and increasing the expression of claudin and ZO-1 proteins, thus preserving the physical barrier function. Moreover, the gut microbiota promotes the synthesis of short-chain fatty acids (SCFAs), which further enhances tight junctions between epithelial cells and alleviates intestinal damage [10,11,12]. Furthermore, the gut microbiota stimulates the immune system by inducing the differentiation of CD4+ T cells into Th17 cells. According to research, changes in the gut microbiota affect tryptophan metabolism in piglets, leading to increased levels of 3-indoleacetic acid and 3-indolepropionic acid in the intestine [13]. Among these, indolepropionic acid activates the aryl hydrocarbon receptor on Th17 cells, promoting their conversion into regulatory T cells (Tregs) [14]. Additionally, the gut microbiota also promotes non-specific immune responses. For example, when recipient piglets were administered fecal suspensions from Jinhua pigs daily, their gut microbial structure changed significantly. On day 12, toll-like receptor 2 (TLR2) and TLR4 levels in the colon were significantly elevated, along with a significant increase in secretory immunoglobulin A (SIgA) in the colon [15,16,17]. Therefore, understanding the influence of the Ningxiang pig microbiota on nutrient absorption processes becomes a focal point of research.

In this study, we focused on the Ningxiang pig, a protected breed of livestock genetic resources in China, as the primary research subject. We analyzed the four main sites of nutrient absorption (stomach, ileum, cecum, and colon) using methods such as 16S rRNA sequencing, OTU identification, alpha diversity analysis, beta diversity analysis, gut microbiota function prediction, and KEGG (Kyoto Encyclopedia of Genes and Genomes) metabolic pathway analysis. Our findings revealed significant differences in the microbial communities across these four key nutrient absorption sites, with enrichment of pathways closely related to nutrient absorption, including cell cycle control and cell division, lipid transport and metabolism, carbohydrate transport and metabolism, and RNA processing and modification. By elucidating the relationships and specific distributions of microbial communities in the main nutrient absorption sites of Ningxiang pigs, we established a strong link between microbiota and nutrient absorption. The aim of this study is to clarify the mechanisms and differences in the role of microbial communities during nutrient absorption in Ningxiang pigs, providing valuable insights for future nutritional regulation strategies. This will guide the application of microbiota transplantation technology and contribute to improving the efficiency of nutrient absorption in Ningxiang pigs, addressing the limitations posed by their slow growth both domestically and internationally.

## 2. Materials and Methods

### 2.1. Ethics Statement

This study was conducted in accordance with the ethical principles set by The Journal of Animals. The whole experimental procedure was strictly carried out by the Biomedical Research Ethics Committee of Hunan Normal University, Changsha, Hunan, China (2018-056, Approval Date: 1 January 2018).

### 2.2. Experimental Animals and Gut Tissues Storage

The Ningxiang pig experiment was conducted by Hunan Chuweixiang Agriculture and Animal Husbandry Co., Ltd. (Ningxiang, Changsha, China). A total of 6 six-month-old Ningxiang pigs with an average weight of 73.41 ± 2.19 kg (3 males and 3 females) were randomly assigned to the experimental groups and fed a basal diet. Throughout the experiment, all pigs were managed under identical conditions. At the end of the trial, the six pigs were humanely euthanized, and samples of gastric chyme (NFG), ileal chyme (NFI), cecal chyme (NFC), and colonic chyme (NFL) were rapidly collected. These samples were then subjected to 16S rRNA sequencing analysis to investigate the microbial community structure and functional characteristics in different segments of the gastrointestinal tract.

### 2.3. 16S rRNA Library Construction

Collect samples from pig gut contents and store them immediately at −80 °C to prevent DNA degradation. Use appropriate lysis buffers and proteinase K to lyse gut microbial cells and release DNA. Remove proteins and other impurities through centrifugation or filtration to obtain high-quality total DNA. Select primers targeting the conserved regions of the 16S rRNA gene, such as 341F (5′-CCTACGGGNGGCWGCAG-3′) and 806R (5′-GGACTACHVGGGTWTCTAAT-3′). Each PCR reaction contains template DNA (about 10–50 ng), forward and reverse primers (each 0.2 μM), dNTPs (0.2 mM), Taq polymerase (1 U), and an appropriate buffer. A typical PCR program includes an initial denaturation step at 95 °C for 5 min, followed by 30–35 cycles of denaturation at 95 °C for 30 s, annealing at 55–60 °C for 30 s, and extension at 72 °C for 60 s, with a final extension step at 72 °C for 10 min. After PCR amplification, purify the PCR products using magnetic beads (such as AMPure XP beads) to remove unincorporated primers and small molecular byproducts. Add sequencing platform-specific adapters and sample-specific barcodes (indices) to both ends of the PCR products to facilitate multiplex sequencing. Perform quality control steps, including quantifying the library using fluorometric methods (e.g., Qubit) and assessing the library fragment size distribution and purity using capillary electrophoresis (e.g., Agilent Bioanalyzer, Santa Clara, CA, USA). Finally, pool the libraries from different samples in proportion and load the pooled library onto a high-throughput sequencing platform, such as Illumina MiSeq or NovaSeq, for sequencing. The 16S rRNA sequencing data have been uploaded to (https://figshare.com/articles/dataset/16s_rRNA_data_of_Ningxiang_pigs/28397135/1, accessed on 1 January 2025).

### 2.4. 16S rRNA Sequencing Data Preprocessing

Raw sequencing data were first subjected to quality filtering using Trimmomatic (version 0.33) [18]. This step involved trimming low-quality bases from the ends of reads and removing reads shorter than a specified length threshold (e.g., 35 bp). Additionally, adapter sequences were trimmed to ensure high-quality reads. Primer sequences were identified and removed using Cutadapt (version 1.9.1) [19]. This step ensured that only the target regions of interest remained in the dataset, thereby reducing noise and improving downstream analysis accuracy. Paired-end reads were merged into full-length sequences using USEARCH (version 10) [20]. During this process, UCHIME (version 8.1) [21] was employed to detect and remove chimeric sequences. After these preprocessing steps, high-quality sequences were obtained and prepared for downstream analysis. These sequences were used for further analyses, such as OTU clustering, taxonomic classification, and diversity assessment.

### 2.5. Species Annotation Methods

Import high-quality feature sequences into the QIIME2 workflow [22]. Use classify-consensus-blast to perform initial alignment and generate preliminary classification results based on the set parameters (e.g., sequence similarity, coverage, and minimum consensus). Check if the initial alignment results meet the preset thresholds. If not, mark these sequences for further classification. For sequences that do not meet the thresholds, use classify-sklearn for classification. The confidence threshold is set to 0.7.

### 2.6. Alpha Diversity Analysis and Beta Diversity Analysis

Using QIIME2 software (v2.0.6), indices such as ACE, Chao1, Simpson, and Shannon were calculated to assess microbial richness and evenness across different gut tissues (e.g., gastric chyme NFG, ileal chyme NFI, cecal chyme NFC, and colonic chyme NFL). To compare differences between groups, alpha diversity analysis was performed using ANOVA, with a significance threshold set at *p* < 0.05. This approach allows us to determine whether there are significant differences in microbial community structure among different tissue types. Species diversity matrices were generated using algorithms like Binary Jaccard, Bray–Curtis, and (un)weighted UniFrac. Principal Coordinates Analysis (PCoA) and Redundancy Analysis (RDA/CCA) were performed using the R language platform [23] to explore correlations between gut microbiota factors and gut group composition.

### 2.7. Significance Analysis of Intergroup Differences

We used LefSe analysis (http://huttenhower.sph.harvard.edu/lefse/, accessed on 1 February 2025) [24] to identify significantly different gut microbial communities between groups. To ensure the reliability of the results, the significant microbial communities were required to meet both of the following criteria: an LDA score greater than or equal to 4.0 and a *p*-value less than 0.05. The LDA score reflects the effect size of abundance differences between groups, while the *p*-value assesses the statistical significance of these differences.

### 2.8. PICRUSt2 Function Prediction

We used PICRUST (v.1.1.0, http://picrust.github.io/picrust/, accessed on 1 February 2025) [25] to predict the potential molecular functions of the bacterial flora within the samples. This approach not only helps us understand the potential roles of microbial communities in key metabolic pathways like amino acid metabolism and carbohydrate metabolism but also reveals their important functions in signaling and other biological processes.

### 2.9. Statistical Analysis

Statistical analyses were performed using the SPSS 18.0 software package (SPSS Science, Chicago, IL, USA). Experimental data were subjected to *t*-test and ANOVA analyses, with a significance threshold set at *p* < 0.05. Graphs were generated using GraphPad Prism 8 software (GraphPad, Santiago, MN, USA). Data were presented as mean ± standard deviation (SD).

## 3. Results

### 3.1. Microbial Community Composition and Diversity Across Nutrient Absorption Tissues

After sequencing the 24 samples and identifying them through barcodes, a total of 299,952 CCS sequences were obtained. Each sample generated at least 7991 CCS sequences, with an average of 12,498 CCS sequences per sample (Appendix A). OTUs (Operational Taxonomic Units) were generated by clustering sequences at a 97% sequence similarity threshold. Each cluster and its representative sequence (referred to as an OTU) were used for statistical analysis of sequence abundance in downstream analyses. A total of 801 OTUs were identified across the 24 samples (Figure 1A). The NFI group exhibited the lowest OTU richness, with an average of 137.2 ± 56.35 (mean ± standard deviation), while the NFL group showed the highest richness, with an average of 506.5 ± 57.88. Significant differences in OTU distribution were observed among the nutrient digestion tissues. Cluster analysis revealed that the NFL group contained 32 unique OTUs, whereas the NFG group had 37 unique OTUs (Figure 1B). Notably, the NFL and NFC groups shared the largest number of overlapping OTUs, approximately 684, which accounted for over 90% of the total OTUs within these two groups. These results demonstrate that the distinct differences in microbial composition among nutrient absorption tissues (such as gastric chyme NFG, ileal chyme NFI, cecal chyme NFC, and colonic chyme NFL) underscore the scientific significance of this study. By revealing these differences, we can better understand the role of gut microbiota in nutrient absorption and overall health.

### 3.2. Sequencing Saturation and Alpha Diversity of Microbial Profiles

We randomly subsampled a fixed number of sequences from each sample and quantified the species represented by these sequences. By plotting the relationship between sequence count and species richness, we evaluated whether the samples had reached saturation. The results demonstrated that the rarefaction curve, Shannon index curve, and species accumulation curve all approached stabilization, indicating sufficient sequencing depth for reliable analysis (Appendix A).

Alpha diversity, a key metric in community ecology, reflects both species richness and evenness. Using alpha diversity indices (ACE, Chao1, Simpson, and Shannon), we identified significant differences among nutrient absorption tissues (Figure 2A–D). Specifically, pairwise comparisons revealed significant disparities in these indices between NFC vs. NFG and NFG vs. NFL.

### 3.3. Microbial Composition and Abundance Profiles in NFC and NFL Reveal Functional Similarities

By aligning the representative sequences of OTUs with the mainstream SILVA database, we obtained species annotations for all OTUs in each sample. At the phylum level, we found that *Firmicutes* dominated the microbial composition, while in NFC and NFL, *Bacteroidota* also constituted a major microbial group, accounting for approximately 15–20% (Figure 3A). On the other hand, at the order level, we observed similar high abundances of certain microbial groups in NFC and NFL, such as *Burkholderiales*, *Bacteroidales*, *Fibrobacterales*, and *Peptococcales* (Figure 3B). The above findings suggest that the microbial communities in NFC and NFL exhibit similar abundance profiles and may perform similar functions in nutrient absorption.

### 3.4. Beta Diversity of Microbial Profiles

Beta diversity is used to measure the differences in species composition between different microbial communities. To investigate the heterogeneity of microbial communities in the four major nutrient absorption tissues (NFC, NFI, NFL, and NFG), we employed principal coordinates analysis (PCoA), correlation analysis between microbial abundance and sample characteristics, as well as analysis of similarities (ANOSIM). The PCoA and correlation analysis between microbial abundance and sample characteristics revealed that the heterogeneity between NFG and NFI was relatively high, while NFC and NFL exhibited heterogeneity but with less distinct separation compared with NFG and NFI (Figure 4A,B). ANOSIM, a statistical method for evaluating similarities among multidimensional datasets, can provide a statistical test for the significance of differences between sample groups observed in PCoA analysis. The results showed that the heterogeneity among NFC, NFL, NFG, and NFI was significant (*p* < 0.05) (Figure 4C), providing strong support for our study of tissue-specific microbial communities associated with nutrient absorption.

### 3.5. LEfSe Analysis Reveals Key Microbial Biomarkers in Nutrient Absorption Tissues

Linear Discriminant Analysis Effect Size (LEfSe) can be used to compare two or more groups and identify statistically significant biomarkers that differ between groups. LEfSe analysis of the microbial community with criteria of *p* < 0.05 and LDA > 4. Through the LDA value distribution bar chart, the dominant bacterial communities in each nutrient absorption tissue can be intuitively observed (Figure 5A and Appendix A). Additionally, by constructing evolutionary branch diagrams using LEfSe analysis (Figure 5B), the phylogenetic information of microbial communities can be explored, allowing for clear and direct visualization of the attributes of different microbial groups. This facilitates a more straightforward presentation of the characteristics of various microbial populations. The dominant microbial groups in NFC are primarily f_*Prevotellaceae*, f_p_*251_o5*, and g_*Agathobacter*. The dominant microbial group in NFG is mainly o_*Lactobacillales* and s_*Lactobacillus_johnsonii*. The dominant microbial groups are NFI is primarily f_*Clostridiaceae*, f_*Peptostreptococcaceae*, and g_*Clostridium_subgroup_stricto_1*. The dominant microbial groups in NFL are mainly f_*Muribaculaceae*, f_*Streptococcaceae*, and g_*Streptococcus* (Figure 5C). These dominant microbial communities play important biological roles in the nutrient absorption of their respective tissues. Therefore, gaining a deeper understanding of the biological characteristics of these microorganisms involved is an essential focus of our research and an area of future attention.

### 3.6. Functional Profiling and Comparative Metabolic Analysis of Microbial Communities Across Nutrient Absorption Tissues

We performed a functional KEGG pathway analysis using PICRUSt2 for the dominant microbial communities at the order level and found that these microorganisms are primarily involved in several key areas: organismal systems (aging, digestive system), cellular processes (cell growth and death, transport, and catabolism), environmental information processing (signaling), genetic information processing (transcription, translation), and metabolic regulation (amino acid metabolism, carbohydrate metabolism) (Figure 6A). To gain a deeper understanding of the functional differences in microbial communities across different nutrient absorption tissues, we conducted a comparative analysis of metabolic abundance between groups. The results showed: In the comparison of NFC vs. NFI, significant differences were observed in the immune system, amino acid metabolism, carbohydrate metabolism, and other pathways (Figure 6B). In the comparison of NFG vs. NFC, significant differences were found in cellular motility, cell growth and death, energy metabolism, amino acid metabolism, and other processes (Figure 6C). In the comparison of NFL vs. NFI, significant differences were identified in transcriptional regulation, signal transduction, carbohydrate metabolism, lipid metabolism, and other pathways (Figure 6D). In the comparison of NFG vs. NFL, significant differences were observed in translational regulation, carbohydrate metabolism, aging, cell growth and death, replication and repair, and other processes (Figure 6E).

Based on the differential metabolic regulation information obtained from these gut nutrient absorption tissues, we can confidently conclude that gut-specific microbial communities play crucial roles in regulating distinct biological functions. These functions are essential for the growth, development, and nutrient absorption of Ningxiang pigs, indicating that gut microbiota are indispensable for maintaining host health and physiological balance.

## 4. Discussion

### 4.1. Analysis of α and β Diversity of Microbial Communities

Gut microbiota plays a crucial role in animal health by not only digesting food (especially important for the digestion of coarse fibers) but also enhancing the activity of intestinal digestive enzymes [26,27]. Additionally, it synthesizes vitamins and other nutrients from the contents of the intestinal lumen, which can be absorbed and utilized by the host organism [28]. The analysis of microbial OTU sequence features revealed that while the four main nutrient absorption tissues (gastric chyme NFG, ileal chyme NFI, cecal chyme NFC, and colonic chyme NFL) in Ningxiang pigs share 319 OTUs, an additional 365 OTUs are shared between NFC (cecal chyme) and NFL (colonic chyme), with these additional shared OTUs predominantly belonging to the genus *Bacteroides*. On the one hand, this indicates significant differences in the number of microbial features among different nutrient absorption sites in the gastrointestinal tract of Ningxiang pigs; on the other hand, it suggests that the microbial communities in the gastrointestinal tract may have distinct functional roles. Considering the gastric–ileal–cecal–colonic food digestion axis, the microbial communities in the cecum and colon typically focus on fermenting amino acid derivatives, playing a crucial role in nutrient absorption and metabolism [29]. Our findings show that the OTU communities in the cecum and colon are predominantly composed of *Bacteroides*, which play a key role in enhancing the host’s resilience and maintaining gut health. Therefore, future research should focus on the regulatory mechanisms between the cecal–colonic and gastric–ileal microbial communities to better understand their roles in nutrient absorption and overall health.

We found that the alpha diversity index values for NFI (ileal chyme) were the lowest, which is consistent with the number of OTUs detected in NFI. This result was surprising because the primary function of the ileum is to absorb nutrients from food, particularly fats and proteins. Through the action of villi, the ileum increases the surface area in contact with food, thereby enhancing nutrient absorption efficiency. We hypothesize that the lower microbial diversity in the ileum may be related to the slower growth and development of Ningxiang pigs [29]. Beta diversity PCoA analysis revealed a clear distinction between NFI and NFG (gastric chyme), while the separation between NFL (colonic chyme) and NFC (cecal chyme) was less pronounced. Additionally, ANOSIM analysis indicated significant differences among the four gastrointestinal microbial communities in Ningxiang pigs (*p* < 0.05). Both alpha and beta diversity analyses have revealed differences in the gut microbial communities [30], which will aid further research into this phenomenon and help uncover key information about these microbial communities.

### 4.2. Comparative Analysis of Microbial Community Composition Across Nutrient Tissues

The analysis of gut microbiota composition showed that at the phylum level, the dominant bacterial phyla in the NFG (gastric chyme) and NFI (ileal chyme) groups were *Firmicutes* (80%), while the average proportion of *Bacteroidetes* was less than 2%. In contrast, in the NFC (cecal chyme) and NFL (colonic chyme) groups, the dominant phyla were *Firmicutes* (over 60%) and *Bacteroidetes* (over 20%). *Firmicutes* and *Bacteroidetes* are the most abundant microbial phyla in the pig gut, and the ratio between these two phyla is closely related to nutrient absorption and energy metabolism in animals [31,32]. The abnormal imbalance in the ratio of *Firmicutes* to *Bacteroidetes* in the ileum of Ningxiang pigs may be a factor contributing to their poor nutrient absorption and slower biological development.

At the genus level, genera such as *Clostridia_UCG_014* and *Lachnospirales*, which produce short-chain fatty acids (SCFAs), were found to have low abundance in the ileum. Studies have shown that *Clostridium* and *Lachnospira* are involved in the synthesis of propionate and butyrate, both of which play an active role in regulating intestinal growth and development, enhancing nutrient absorption, and improving inflammatory responses and oxidative states [33,34]. Therefore, the low abundance of *Clostridia_UCG_014* and *Lachnospiralesin* the ileum of Ningxiang pigs could be another factor contributing to their slow nutrient absorption. These findings suggest that improving the microbial community environment in the ileum of Ningxiang pigs might be a critical focus for enhancing their growth and development.

### 4.3. Characterization of Tissue-Specific Microbes and Differential Metabolic Pathways in Nutrient Tissues

The non-parametric Kruskal–Wallis (KW) sum-rank test was used to detect differences in species abundance between groups, followed by LEfSe analysis of the microbial community with criteria of *p* < 0.05 and LDA > 4. This allowed us to identify 77 dominant gut microbial taxa, including 22 each for CFC, CFL, and CFG, and only 12 for CFI. Notably, among these 77 dominant gut microbes, there were 12 belonging to p__*Proteobacteria*, 13 to p__*Bacteroidota*, and 52 to p__*Firmicutes* (Appendix A). The preceding discussion indicates that the microbial community in the ileum of Ningxiang pigs may be a key factor contributing to their slower growth and development. An interesting finding is that f__*Peptostreptococcaceae* (family *Peptostreptococcaceae*) is significantly more abundant in the ileum (with a maximum abundance of 0.28 and an average abundance of 0.14), while it is almost undetectable in other tissues (abundance around 0.01). Research has shown that *Peptostreptococcus* is closely associated with regulating intestinal epithelial barrier function, inflammatory responses, bacterial infections, cell proliferation, and indole aromatic hydrocarbon metabolism, but it is often linked to the activation of intestinal inflammation [35,36,37,38]. We have reason to believe that the high abundance of f__*Peptostreptococcaceae* in the ileum may trigger intestinal inflammation during the growth and development of Ningxiang pigs, reduce nutrient absorption efficiency, and thus result in slower growth compared with international pig breeds such as Duroc, Landrace, and Large White.

Through KEGG analysis of these gastrointestinal-specific microbial communities, we particularly focused on the regulatory mechanisms of the microbiota in the ileum of Ningxiang pigs. Our findings indicate that there are significant differences between the ileum and the cecum and colon in terms of lipid metabolism, amino acid metabolism, immune system regulation, gene transcription regulation, and digestive system functions. Research shows that gut microbiota can regulate lipid metabolism and enhance the absorption of lipids in the pig intestine, further promoting the accumulation of lipid droplets in the liver and thereby improving meat quality and flavor [39,40]. Additionally, the nutritional strategies of the gut microbial ecosystem produce a series of amino acid regulations that affect the host environment. For example, tryptophan promotes the growth of beneficial bacteria and inhibits pathogens, while arginine metabolism affects nitrogen cycling, influencing gut immune responses and health. Glutamate and glutamine can increase the levels of beneficial bacteria and reduce the impact of pathogenic bacteria [41]. The immune system and gene transcription are essential factors in the growth and development of Ningxiang pigs [42,43].

This series of pathway regulations suggests that the regulation of the microbiota in the ileum may be a key factor in the growth and development of Ningxiang pigs. Microbiota transplantation could provide an effective solution to this issue, offering support for the growth of Ningxiang pigs. However, our current research is still in its preliminary stages. By conducting further studies on individual differences and dietary habits and integrating livestock and dietary metadata, we can gain a more comprehensive understanding of the relationship between the microbiome and animal health and production performance.

## 5. Conclusions

The microbial community in the ileum of Ningxiang pigs appears to be a critical factor affecting their growth and development. The observed imbalances in microbial diversity and composition, particularly the high abundance of *Peptostreptococcaceae* and the low abundance of beneficial SCFA-producing genera, likely contribute to reduced nutrient absorption and increased inflammation. Addressing these issues through microbiota transplantation and further research into dietary and individual factors holds promise for improving the growth and health of Ningxiang pigs. Future work should aim to validate these findings and explore practical applications, such as optimizing feed formulations and developing targeted probiotic interventions, to enhance the productivity and well-being of Ningxiang pigs.

## Figures and Tables

**Figure 1 animals-15-00936-f001:**
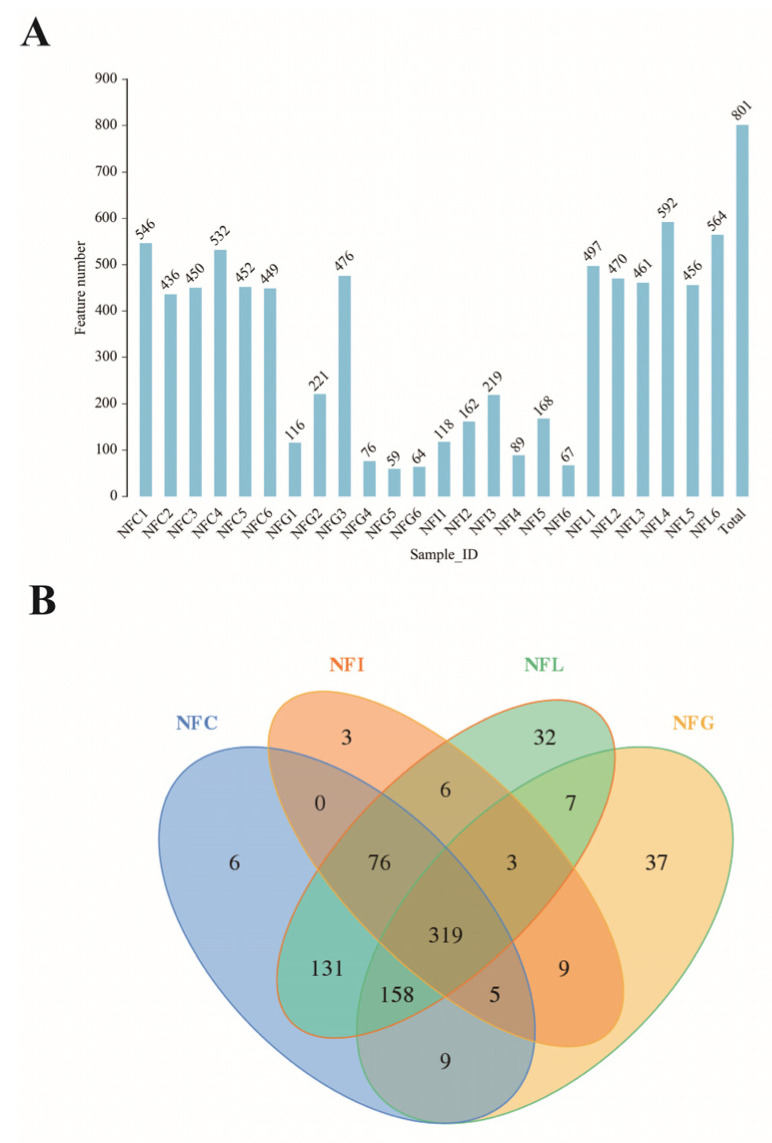
**OTU-based microbial community composition.** (**A**): Distribution of OTU counts in samples. (**B**): Venn diagram illustrating group-level analysis of OTU distribution. NFC: cecal content, NFI: ileal content, NFL: colonic content, NFG: gastric content.

**Figure 2 animals-15-00936-f002:**
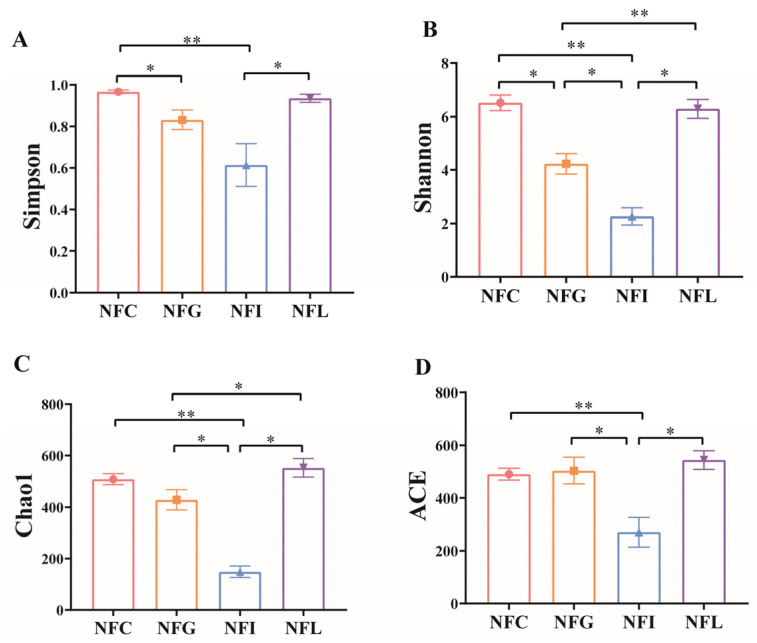
**Alpha diversity of microbial profiles.** These show the (**A**) Simpson index, (**B**) Shannon index, (**C**) Chao1 index, and (**D**) ACE index, respectively. The bar graphs with * on top indicate statistical significance (*p* < 0.05), and ** on top indicates statistical significance (*p* < 0.01).

**Figure 3 animals-15-00936-f003:**
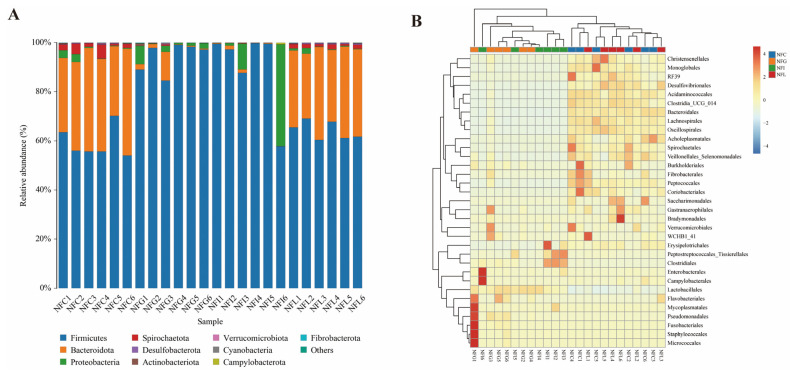
**Microbial composition and abundance profiles.** (**A**): Gut microbiota profiling at the phylum level. (**B**): Heatmap profiling of gut microbiota at the order level.

**Figure 4 animals-15-00936-f004:**
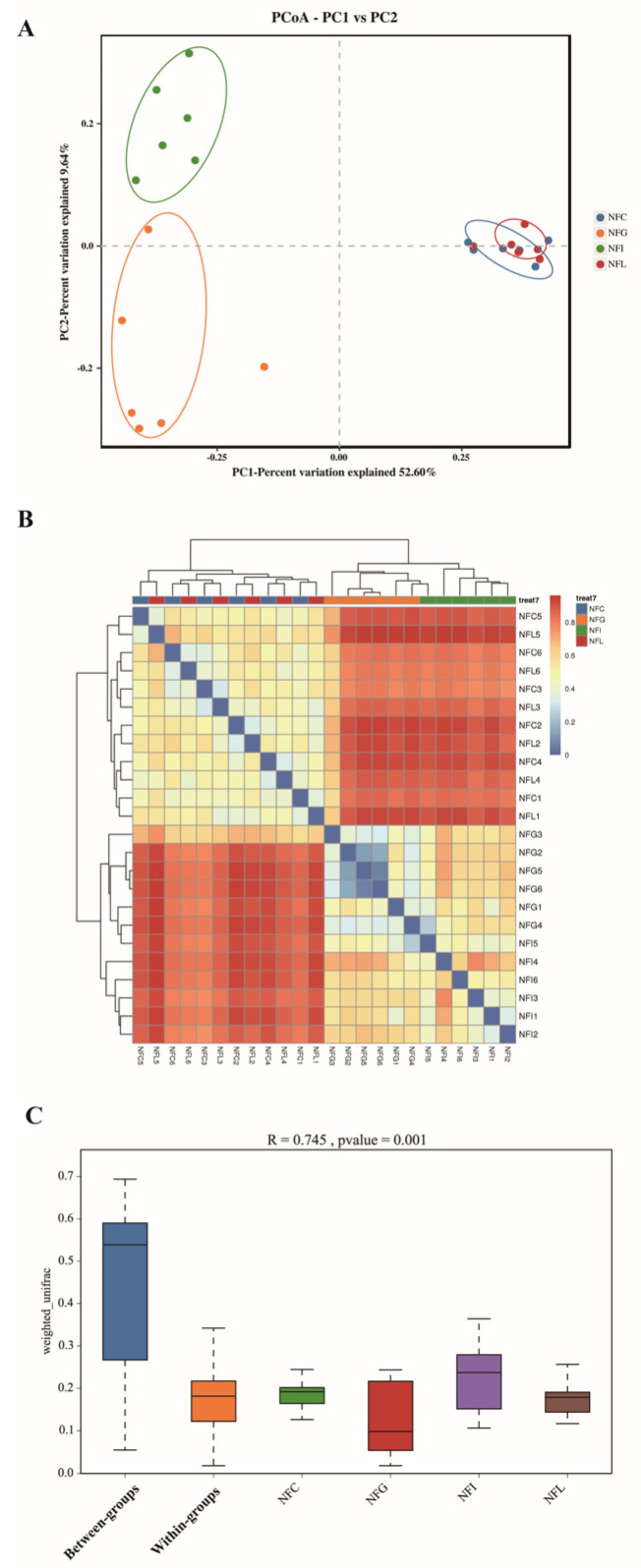
**Beta diversity of microbial profiles.** (**A**): PCoA score plot. (**B**): Correlation analysis based on microbial abundance to determine the relationships between samples. (**C**): Anosim analysis for beta diversity between different sample groups.

**Figure 5 animals-15-00936-f005:**
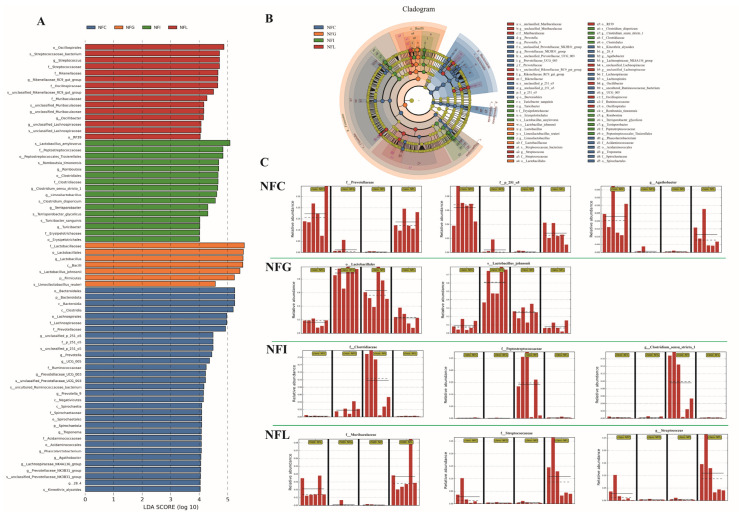
**LEfSe analysis reveals key microbial biomarkers.** (**A**): LDA score of gut microbiota by LEfSe analysis. (**B**): Taxonomic cladogram obtained from linear discriminant analysis effect size (LEfSe) sequence analysis. Species taxonomy from phyla to genus (inside–outside). The diameter of each circle represents the relative abundance of the taxon, and the color corresponds to the grouping. (**C**): Gut-specific high-expression microbial abundance distribution. The relative abundance of bacterial genera significantly recovered (*p* < 0.05); solid and dashed lines indicate the mean and median, respectively. NFC: cecal content, NFI: ileal content, NFL: colonic content, NFG: gastric content.

**Figure 6 animals-15-00936-f006:**
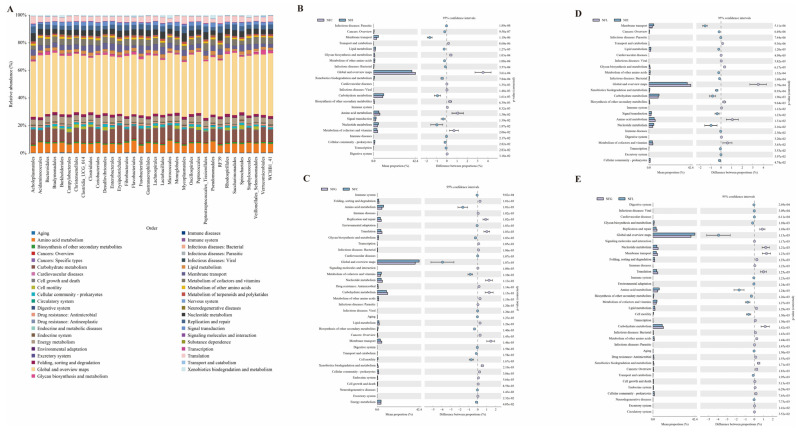
**Functional profiling and comparative metabolic analysis of microbial communities.** (**A**): KEGG pathways of functional genes for microbial communities. (**B**–**E**): Differential metabolic functional information between sample gut groups.

## Data Availability

https://figshare.com/articles/dataset/16s_rRNA_data_of_Ningxiang_pigs/28397135/1, accessed on 1 February 2025.

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
