# Peer review of "Comparative Analysis of Gut Microbiota Diversity Across Different Digestive Tract Sites in Ningxiang Pigs"

_animals, 2025, doi:10.3390/ani15070936_

Round 1
Reviewer 1 Report
Comments and Suggestions for Authors
In this paper, 16S rRNA sequencing was performed on the content samples of four nutrient absorption sites of Ningxiang pigs, the microbial community composition and species abundance were analyzed, and the function was analyzed. However, the Materials and Methods section lacks detailed experimental procedures. Additionally, the Discussion section provides an excessive description of the results with limited analysis, making it difficult to establish the relationship between microbial communities in different intestinal segments and nutrient absorption. Furthermore, the analysis of differential microbiota and functional differences has not been fully explored.
Introduction
- Line 62 (Page 2): Please provide more detailed information on how research into nutrient absorption contributes to improving economic efficiency, and further elaborate on the significance of nutrient absorption studies specifically for Ningxiang pigs.
- Line 69 (Page 2): Italics are required for species and genera, "Prevotella" and "Bacteroides" should be italicized.
- Line 75 (Page 2): The citation should be changed to [10-12].
- Line 85 (Page 2): The literature citation should be changed to [15-17].
- Line 92 (Page 2): Provide the full form of KEGG the first time it is mentioned.
Materials and Methods:
- The authors describe many methodological definitions but lack specific details on the analytical methods and steps involved. Please provide a more detailed description of the methods used.
- Line 112 (Page 3): The article only provided the weight information of animals. Please provide additional information about the experimental animals, such as their sex and age.
- Line116 (Page 3): Please add the relevant details about sample collection, DNA extraction, PCR amplification and other experimental steps.
- Line 119 (Page 3): The title is incorrectly written; the "." after "16"should be removed.
- Line 179 (Page 4): Please specify the analysis methods used for each data set. For example, "Two-tailed Wilcoxon rank-sum tests were used to analyze the ACE and Shannon indices."
Results
- Line 200-203 (Page 4): The overlap of OTUs alone does not necessarily indicate a strong relationship between microbiotain different regions. It is recommended to remove this sentence.
- Line 221 (Page 6): The grouping in the study is designated as NFC, NFG, NFI, and NFL; however, in Figure 2 are presented as CFC, CFG, CFI, and CFL.
- Line 277 (Page 9):I suggest improving the quality of the 5 figure on page 9.
- Line 295 (Page 10): Why is there no comparison between NFG vs NFI and NFC vs NFL? The PCoA distance between these groups is closer, or because there is no significant difference in the comparisons?
Discussion
- Line 313 (Page 10): The Discussion section is overly focused on presenting the results, with limited meaningful analysis. There is no clear connection established between themicrobiota and nutrient absorption in each intestinal segment, and the analysis of differential microbiota and functions has not been fully explored.
- Line 361-365 (Page 11):The grouping in the study is designated as NFC, NFG, NFI, and NFL, but this paragraph mentions CFC, CFG, CFI, and CFL, which is the same as the error in Figure 2. Please ensure the group names are consistent throughout the paper.
The language of this manuscript could be improved. Some sentences are unclear or overly complex, which may affect the readability and flow of the paper.
Author Response
Thank you very much for your valuable feedback. Your suggestions have greatly improved the research prospects and overall quality of our manuscript. We have carefully addressed each of your comments and have made the necessary revisions. Our detailed responses to your feedback are provided in the attached document. We appreciate your attention to this matter.

Reviewer 2 Report
Comments and Suggestions for Authors
The main question addressed is about tissue-specific nutritional absorption comparison between different body sites of Ningxiang pigs based on differences in microbial communities in tissues. This is a very unique approach.
To the best of my knowledge, this topic is very original and relevant. Previously, many studies have been done on microbiota, and almost always, there is a predominance of Proteobacteria, whatever the sample is. There are also other research articles that studied differential gene function using KEGG pathways, but nobody has ever discussed how microbial genes could be accessed to see their contribution toward nutrient absorption in metabolic pathways. This is significant as it tells us which microbial community is more efficient, whether transplants could be done, and which bacteria are more efficient in metabolic pathways.
Other publications I have read so far either focus on metagenomics or end with diversity indices, but this research applies further tools to explore the functional profiling of the microbiome in different tissues of Ningxiang pigs.
To improve the methodology, the author can add more details on data handling, including the tools used, their outputs, and the input file types of each software used in this study.
The references are up-to-date and appropriate.
The tables and figures are of good quality and satisfactory.
Overall the manuscript is well structured and comprehensive work has been done. However, conclusion of the study needs to be revised, currently its more about methodology and aims rather than what actually has been concluded from the study. I would recommend the author to emphasize on insights gained from the results including broader application of the findings.
Author Response

(The authors gave the same response as above.)

Reviewer 3 Report
Comments and Suggestions for Authors
The manuscript describes the comparison of the microbiome in different parts of the pig digestive tract. The conducted studies are an important addition to the information on the microbiome of the digestive tract, as in pigs most studies concern the microbiome of feces or the microbiome of the lower digestive tract. The work is written in an interesting way and contains many analyses of the microbiome, My suggestions:
1) I suggest changing the title. The title is misleading because it suggests that different organs will be studied, such as lungs, vagina and not different parts of the digestive tract. Moreover, metabolic function is also misleading and should be removed
2) there are more revelent references for the effect of microbiome on growth in pgs, for example: Maltecca C, Bergamaschi M, Tiezzi F. The interaction between microbiome and pig efficiency: A review. J Anim Breed Genet. 2020 Jan;137(1):4-13. doi: 10.1111/jbg.12443. Epub 2019 Oct 1. PMID: 31576623.
lines 107-109 Please provide the number of Ethical commitee statement
lines 116-117 Please provide the details of 16sRNA library construction
lines 126-178 this fragments sound like the part of the report from sequencing company
lines 257-266 as above
line 314 Nutrient tissues?
line 323 Please rewrite the sentence (it is obvious that different parts of GI play different roles)
Conclusions could be shortened
Supplementary figure 1 suggest that increasing sequencing depth could improve the outpu data
Author Response

(The authors gave the same response as above.)

Round 2
Reviewer 1 Report
Comments and Suggestions for Authors
The author added relevant information in the methodology section, improved the clarity of the presentation of the results, and restructured the discussion. The manuscript has been significantly improved upon the first submission. However, a few minor corrections remain to be addressed:Species and genus names need italics, while other taxonomic mediators do not need italics. Please check the manuscript.
Author Response

(The authors gave the same response as above.)
